# Regret based Robust Solutions for Uncertain Markov Decision Processes

**Asrar Ahmed**
Singapore Management University
masrara@smu.edu.sg

**Pradeep Varakantham**
Singapore Management University
pradeepv@smu.edu.sg

**Yossiri Adulyasak**
Massachusetts Institute of Technology
yossiri@smart.mit.edu

**Patrick Jaillet**
Massachusetts Institute of Technology
jaillet@mit.edu

## Abstract

In this paper, we seek robust policies for uncertain Markov Decision Processes (MDPs). Most robust optimization approaches for these problems have focussed on the computation of *maximin* policies which maximize the value corresponding to the worst realization of the uncertainty. Recent work has proposed *minimax* regret as a suitable alternative to the *maximin* objective for robust optimization. However, existing algorithms for handling *minimax* regret are restricted to models with uncertainty over rewards only. We provide algorithms that employ sampling to improve across multiple dimensions: (a) Handle uncertainties over both transition and reward models; (b) Dependence of model uncertainties across state, action pairs and decision epochs; (c) Scalability and quality bounds. Finally, to demonstrate the empirical effectiveness of our sampling approaches, we provide comparisons against benchmark algorithms on two domains from literature. We also provide a Sample Average Approximation (SAA) analysis to compute a posteriori error bounds.

## Introduction

Motivated by the difficulty in exact specification of reward and transition models, researchers have proposed the uncertain Markov Decision Process (MDP) model and robustness objectives in solving these models. Given the uncertainty over the reward and transition models, a robust solution can typically provide some guarantees on the worst case performance. Most of the research in computing robust solutions has assumed a *maximin* objective, where one computes a policy that maximizes the value corresponding to the worst case realization [8, 4, 3, 1, 7]. This line of work has developed scalable algorithms by exploiting independence of uncertainties across states and convexity of uncertainty sets. Recently, techniques have been proposed to deal with dependence of uncertainties [15, 6].

Regan *et al.* [11] and Xu *et al.* [16] have proposed *minimax* regret criterion [13] as a suitable alternative to *maximin* objective for uncertain MDPs. We also focus on this *minimax* notion of robustness and also provide a new myopic variant of regret called Cumulative Expected Regret (CER) that allows for development of scalable algorithms.

Due to the complexity of computing optimal *minimax* regret policies [16] , existing algorithms [12] are restricted to handling uncertainty only in reward models and the uncertainties are independent across states. Recent research has shown that sampling-based techniques [5, 9] are not only efficient but also provide a priori (Chernoff-Hoeffding bounds) and a posteriori [14] quality bounds for planning under uncertainty.

In this paper, we also employ sampling-based approaches to address restrictions of existing approaches for obtaining regret-based solutions for uncertain MDPs . More specifically, we make the

following contributions: (a) An approximate Mixed Integer Linear Programming (MILP) formulation with error bounds for computing minimum regret solutions for uncertain MDPs, where the uncertainties across states are dependent. We further provide enhancements and error bounds to improve applicability. (b) We introduce a new myopic concept of regret, referred to as Cumulative Expected Regret (CER) that is intuitive and that allows for development of scalable approaches. (c) Finally, we perform a Sample Average Approximation (SAA) analysis to provide experimental bounds for our approaches on benchmark problems from literature.

## Preliminaries

We now formally define the two regret criterion that will be employed in this paper. In the definitions below, we assume an underlying MDP, $\mathcal{M} = \langle \mathcal{S}, \mathcal{A}, T, R, H \rangle$ where a policy is represented as: $\vec{\pi}^t = \{\pi^t, \pi^{t+1}, \ldots, \pi^{H-1}\}$, the optimal policy as $\vec{\pi}^*$ and the optimal expected value as $v^0(\vec{\pi}^*)$. The maximum reward in any state $s$ is denoted as $R^*(s) = \max_a R(s, a)$. Throughout the paper, we use $\alpha(s)$ to denote the starting state distribution in state $s$ and $\gamma$ to represent the discount factor.

**Definition 1** *Regret for any policy $\vec{\pi}^0$ is denoted by $reg(\vec{\pi}^0)$ and is defined as:*

$$reg(\vec{\pi}^0) = v^0(\vec{\pi}^*) - v^0(\vec{\pi}^0), \textbf{where } v^0(\vec{\pi}^0) = \sum_s \alpha(s) \cdot v^0(s, \vec{\pi}^0),$$

$$v^t(s, \vec{\pi}^t) = \sum_a \pi^t(s, a) \cdot \left[ R(s, a) + \gamma \sum_{s'} T(s, a, s') \cdot v^{t+1}(s', \vec{\pi}^{t+1}) \right]$$

Extending the definitions of simple and cumulative regret in stochastic multi-armed bandit problems [2], we now define a new variant of regret called Cumulative Expected Regret (CER).

**Definition 2** *CER for policy $\vec{\pi}^0$ is denoted by $creg(\vec{\pi}^0)$ and is defined as:*

$$creg(\vec{\pi}^0) = \sum_s \alpha(s) \cdot creg^0(s, \vec{\pi}^0), \quad \textbf{where}$$

$$creg^t(s, \vec{\pi}^t) = \sum_a \pi^t(s, a) \cdot \left[ R^*(s) - R(s, a) + \gamma \sum_{s'} T(s, a, s') \cdot creg^{t+1}(s', \vec{\pi}^{t+1}) \right] \quad (1)$$

The following properties highlight the dependencies between regret and CER.

**Proposition 1** *For a policy $\vec{\pi}^0 : 0 \leq reg(\vec{\pi}^0) - creg(\vec{\pi}^0) \leq \left[ \max_s R^*(s) - \min_s R^*(s) \right] \cdot \frac{(1-\gamma^H)}{1-\gamma}$*

**Proof Sketch**[1] By rewriting Equation (1) as $creg(\vec{\pi}^0) = v^{0,\#}(\vec{\pi}^0) - v^0(\vec{\pi}^0)$, we provide the proof.

**Corollary 1** *If $\forall s, s' \in \mathcal{S} : R^*(s) = R^*(s')$, then $\forall \vec{\pi}^0 : creg(\vec{\pi}^0) = reg(\vec{\pi}^0)$.*

**Proof.** Substituting $\max_s R^*(s) = \min_s R^*(s)$ in the result of Proposition 1, we have $creg(\vec{\pi}^0) = reg(\vec{\pi}^0)$. ∎

## Uncertain MDP

A finite horizon uncertain MDP is defined as the tuple of $\langle \mathcal{S}, \mathcal{A}, \mathbf{T}, \mathbf{R}, H \rangle$. $\mathcal{S}$ denotes the set of states and $\mathcal{A}$ denotes the set of actions. $\mathbf{T} = \Delta^\tau(\mathcal{T})$ denotes a distribution over the set of transition functions $\mathcal{T}$, where $\mathcal{T}_k^t(s, a, s')$ denotes the probability of transitioning from state $s \in \mathcal{S}$ to state $s' \in \mathcal{S}$ on taking action $a \in \mathcal{A}$ at time step $t$ according to the $k$th element in $\mathcal{T}$. Similarly, $\mathbf{R} = \Delta^\rho(\mathcal{R})$ denotes the distribution over the set of reward functions $\mathcal{R}$, where $\mathcal{R}_k^t(s, a, s')$ is the reinforcement obtained on taking action $a$ in state $s$ and transitioning to state $s'$ at time $t$ according to $k$th element in $\mathcal{R}$. Both $\mathcal{T}$ and $\mathcal{R}$ sets can have infinite elements. Finally, $H$ is the time horizon.

In the above representation, every element of $\mathcal{T}$ and $\mathcal{R}$ represent uncertainty over the entire horizon and hence this representation captures dependence in uncertainty distributions across states. We now provide a formal definition for the independence of uncertainty distributions that is equivalent to the rectangularity property introduced in Iyengar *et al.* [4].

**Definition 3** *An uncertainty distribution $\Delta^\tau$ over the set of transition functions, $\mathcal{T}$ is independent over state-action pairs at various decision epochs if*

$$\Delta^\tau(\mathcal{T}) = \times_{s \in \mathcal{S}, a \in \mathcal{A}, t \leq H} \Delta_{s,a}^{\tau,t}(\mathcal{T}_{s,a}^t), \ i.e. \ \forall k, Pr_{\Delta^\tau}(\mathcal{T}^k) = \prod_{s,a,t} Pr_{\Delta_{s,a}^{\tau,t}}(\mathcal{T}_{s,a}^t)$$

*where $\mathcal{T} = \times_{s,a,t} \mathcal{T}_{s,a}^t$, $\mathcal{T}_{s,a}^t$ is the set of transition functions for $s, a, t$; $\Delta_{s,a}^{\tau,t}$ is the distribution over the set $\mathcal{T}_{s,a}^t$ and $Pr_{\Delta^\tau}(\mathcal{T}^k)$ is the probability of the transition function $\mathcal{T}^k$ given the distribution $\Delta^\tau$.*

We can provide a similar definition for the independence of uncertainty distributions over the reward functions. In the following definitions, we include transition, $T$ and reward, $R$ models as subscripts to indicate value ($v$), regret ($reg$) and CER ($creg$) functions corresponding to a specific MDP. Existing works on computation of *maximin* policies have the following objective:

$$\pi^{maximin} = \arg \max_{\vec{\pi}^0} \min_{T \in \mathcal{T}, R \in \mathcal{R}} \sum_s \alpha(s) \cdot v_{T,R}^0(s, \vec{\pi}^0)$$

Our goal is to compute policies that minimize the maximum regret or cumulative regret over possible models of transitional and reward uncertainty.

$$\pi^{reg} = \arg \min_{\vec{\pi}^0} \max_{T \in \mathcal{T}, R \in \mathcal{R}} reg_{T,R}(\vec{\pi}^0); \pi^{creg} = \arg \min_{\vec{\pi}^0} \max_{T \in \mathcal{T}, R \in \mathcal{R}} creg_{T,R}(\vec{\pi}^0)$$

## Regret Minimizing Solution

We will first consider the more general case of dependent uncertainty distributions. Our approach to obtaining regret minimizing solution relies on sampling the uncertainty distributions over the transition and reward models. We formulate the regret minimization problem over the sample set as an optimization problem and then approximate it as a Mixed Integer Linear Program (MILP).

We now describe the representation of a sample and the definition of optimal expected value for a sample, a key component in the computation of regret. Since there are dependencies amongst uncertainties, we can only sample from $\Delta^\tau$, $\Delta^\rho$ and not from $\Delta_{s,a}^{\tau,t}$, $\Delta_{s,a}^{\rho,t}$. Thus, a sample is:

$$\xi_q = \{\langle \mathcal{T}_q^0, \mathcal{R}_q^0 \rangle, \langle \mathcal{T}_q^1, \mathcal{R}_q^1 \rangle, \cdots \langle \mathcal{T}_q^{H-1}, \mathcal{R}_q^{H-1} \rangle\}$$

where $\mathcal{T}_q^t$ and $\mathcal{R}_q^t$ refer to the transition and reward model respectively at time step $t$ in sample $q$. Let $\vec{\pi}^t$ represent the policy for each time step from $t$ to $H-1$ and the set of samples be denoted by $\xi$. Intuitively, that corresponds to $|\xi|$ number of discrete MDPs and our goal is to compute *one* policy that minimizes the regret over all the $|\xi|$ MDPs, i.e.

$$\pi^{reg} = \arg \min_{\vec{\pi}^0} \max_{\xi_q \in \xi} \sum_s \alpha(s) \cdot [v_{\xi_q}^*(s) - v_{\xi_q}^0(s, \vec{\pi}^0)]$$

where $v_{\xi_q}^*$ and $v_{\xi_q}^0(s, \vec{\pi}^0)$ denote the optimal expected value and expected value for policy $\vec{\pi}^0$ respectively of the sample $\xi_q$.

Let, $\vec{\pi}^0$ be any policy corresponding to the sample $\xi_q$, then the expected value is defined as follows:

$$v_{\xi_q}^t(s, \vec{\pi}^t) = \sum_a \pi^t(s, a) \cdot v_{\xi_q}^t(s, a, \vec{\pi}^t), \textbf{\textit{where }} v_{\xi_q}^t(s, a, \vec{\pi}^t) = \mathcal{R}_q^t(s, a) + \gamma \sum_{s'} v_{\xi_q}^{t+1}(s', \vec{\pi}^{t+1}) \cdot \mathcal{T}_q^t(s, a, s')$$

The optimization problem for computing the regret minimizing policy corresponding to sample set $\xi$ is then defined as follows:

$$\min_{\vec{\pi}^0} \ reg(\vec{\pi}^0)$$

$$\textbf{s.t.} \ \ reg(\vec{\pi}^0) \geq v_{\xi_q}^* - \sum_s \alpha(s) \cdot v_{\xi_q}^0(s, \vec{\pi}^0) \qquad\qquad \forall \xi_q \quad (2)$$

$$v_{\xi_q}^t(s, \vec{\pi}^t) = \sum_a \pi^t(s, a) \cdot v_{\xi_q}^t(s, a, \vec{\pi}^t) \qquad\qquad \forall s, \xi_q, t \quad (3)$$

$$v_{\xi_q}^t(s, a, \vec{\pi}^t) = \mathcal{R}_q^t(s, a) + \gamma \sum_{s'} v_{\xi_q}^{t+1}(s', \vec{\pi}^{t+1}) \cdot \mathcal{T}_q^t(s, a, s') \qquad \forall s, a, \xi_q, t \quad (4)$$

The value function expression in Equation (3) is a product of two variables, $\pi^t(s, a)$ and $v_{\xi_q}^t(s, a, \vec{\pi}^t)$, which hampers scalability significantly. We now linearize these nonlinear terms.

**Mixed Integer Linear Program**

The optimal policy for minimizing maximum regret in the general case is randomized. However, to account for domains which only allow for deterministic policies, we provide linearization separately for the two cases of deterministic and randomized policies.

**Deterministic Policy:** In case of deterministic policies, we replace Equation (3) with the following equivalent integer linear constraints:

$$v_{\xi_q}^t(s, \vec{\pi}^t) \leq v_{\xi_q}^t(s, a, \vec{\pi}^t) \; ; \; v_{\xi_q}^t(s, \vec{\pi}^t) \leq \pi^t(s, a) \cdot M$$
$$v_{\xi_q}^t(s, \vec{\pi}^t) \geq v_{\xi_q}^t(s, a, \vec{\pi}^t) - (1 - \pi^t(s, a)) \cdot M \quad \forall s, a, \xi_q, t \tag{5}$$

$M$ is a large positive constant that is an upper bound on $v_{\xi_q}^t(s, a, \vec{\pi}^t)$. Equivalence to the product terms in Equation (3) can be verified by considering all values of $\pi^t(s, a)$.

**Randomized Policy:** When $\vec{\pi}^0$ is a randomized policy, we have a product of two continuous variables. We provide a mixed integer linear approximation to address the product terms above. Let,

$$A_{\xi_q}^t(s, a, \vec{\pi}^t) = \frac{v_{\xi_q}^t(s, a, \vec{\pi}^t) + \pi^t(s, a)}{2}; B_{\xi_q}^t(s, a, \vec{\pi}^t) = \frac{v_{\xi_q}^t(s, a, \vec{\pi}^t) - \pi^t(s, a)}{2}$$

Equation (3) can then be rewritten as:

$$v_{\xi_q}^t(s, \vec{\pi}^t) = \sum_a [A_{\xi_q}^t(s, a, \vec{\pi}^t)^2 - B_{\xi_q}^t(s, a, \vec{\pi}^t)^2] \tag{6}$$

As discussed in the next subsection on "Pruning dominated actions", we can compute upper and lower bounds for $v_{\xi_q}^t(s, a, \vec{\pi}^t)$ and hence for $A_{\xi_q}^t(s, a, \vec{\pi}^t)$ and $B_{\xi_q}^t(s, a, \vec{\pi}^t)$. We approximate the squared terms by using piecewise linear components that provide an upper bound on the squared terms. We employ a standard method from literature of dividing the variable range into multiple break points. More specifically, we divide the overall range of $A_{\xi_q}^t(s, a, \vec{\pi}^t)$ (or $B_{\xi_q}^t(s, a, \vec{\pi}^t)$), say $[br_0, br_r]$ into $r$ intervals by using $r+1$ points namely $\langle br_0, br_1, \ldots, br_r \rangle$. We associate a linear variable, $\lambda_{\xi_q}^t(s, a, w)$ with each break point $w$ and then approximate $A_{\xi_q}^t(s, a, \vec{\pi}^t)^2$ (and $B_{\xi_q}^t(s, a, \vec{\pi}^t)^2$) as follows:

$$A_{\xi_q}^t(s, a, \vec{\pi}^t) = \sum_w \lambda_{\xi_q}^t(s, a, w) \cdot br_w, \qquad\qquad \forall s, a, \xi_q, t \tag{7}$$

$$A_{\xi_q}^t(s, a, \vec{\pi}^t)^2 = \sum_w \lambda_{\xi_q}^t(s, a, w) \cdot (br_w)^2, \qquad\qquad \forall s, a, \xi_q, t \tag{8}$$

$$\sum_w \lambda_{\xi_q}^t(s, a, w) = 1, \qquad\qquad \forall s, a, \xi_q, t \tag{9}$$

$$SOS2_{\xi_q}^{s,a,t}(\{\lambda_{\xi_q}^t(s, a, w)\}_{w \leq r}), \qquad\qquad \forall s, a, \xi_q, t$$

where $SOS2$ is a construct which is associated with a set of variables of which at most two variables can be non-zero and if two variables are non-zero they must be adjacent. Since any number in the range lies between at most two adjacent points, we have the above constructs for the $\lambda_{\xi_q}^t(s, a, w)$ variables. We implement the above adjacency constraints on $\lambda_{\xi_q}^t(s, a, w)$ using the CPLEX Special Ordered Sets (SOS) type 2[2].

**Proposition 2** *Let [c,d] denote the range of values for $A_{\xi_q}^t(s, a, \vec{\pi}^t)$ and assume we have $r + 1$ points that divide $A_{\xi_q}^t(s, a, \vec{\pi}^t)^2$ into $r$ equal intervals of size $\epsilon = \frac{d^2 - c^2}{r}$ then the approximation error $\delta < \frac{\epsilon}{4}$.*

**Proof:** Let the $r+1$ points be $br_0, \ldots, br_r$. By definition, we have $(br_w)^2 = (br_{w-1})^2 + \epsilon$. Because of the convexity of $x^2$ function, the maximum approximation error in any interval $[br_{w-1}, br_w]$ occurs at its mid-point[3]. Hence, approximation error $\delta$ is given by:

$$\delta \quad \leq \quad \frac{(br_w)^2 + (br_{w-1})^2}{2} - \left[\frac{br_w + br_{w-1}}{2}\right]^2 = \frac{\epsilon + 2 \cdot br_{w-1} \cdot (br_{w-1} - br_w)}{4} < \frac{\epsilon}{4} \quad \blacksquare$$

**Proposition 3** *Let $\hat{v}^t_{\xi_q}(s, \vec{\pi}^t)$ denote the approximation of $v^t_{\xi_q}(s, \vec{\pi}^t)$. Then*

$$v^t_{\xi_q}(s, \vec{\pi}^t) - \frac{|\mathcal{A}| \cdot \epsilon \cdot (1 - \gamma^{H-1})}{4 \cdot (1 - \gamma)} \leq \hat{v}^t_{\xi_q}(s, \vec{\pi}^t) \leq v^t_{\xi_q}(s, \vec{\pi}^t) + \frac{|\mathcal{A}| \cdot \epsilon \cdot (1 - \gamma^{H-1})}{4 \cdot (1 - \gamma)}$$

**Proof Sketch[4]:** We use the approximation error provided in Proposition 2 and propagate it through the value function update. ∎

**Corollary 2** *The positive and negative errors in regret are bounded by* $\frac{|\mathcal{A}| \cdot \epsilon \cdot (1 - \gamma^{H-1})}{4 \cdot (1 - \gamma)}$

**Proof.** From Equation (2) and Proposition 3, we have the proof. ∎

Since the break points are fixed before hand, we can find tighter bounds (refer to Proof of Proposition 2). Also, we can further improve on the performance (both run-time and solution quality) of the MILP by pruning out dominated actions and adopting clever sampling strategies as discussed in the next subsections.

**Pruning dominated actions**

We now introduce a pruning approach[5] to remove actions that will never be assigned a positive probability in a regret minimization strategy. For every state-action pair at each time step, we define a minimum and maximum value function as follows:

$$v^{t,min}_{\xi_q}(s, a) = \mathcal{R}^t_q(s, a) + \gamma \sum_{s'} \mathcal{T}^t_q(s, a, s') \cdot v^{t+1,min}_{\xi_q}(s') \; ; \; v^{t,min}_{\xi_q}(s) = min_a \left\{ v^{t,min}_{\xi_q}(s, a) \right\}$$

$$v^{t,max}_{\xi_q}(s, a) = \mathcal{R}^t_q(s, a) + \gamma \sum_{s'} \mathcal{T}^t_q(s, a, s') \cdot v^{t+1,max}_{\xi_q}(s') \; ; \; v^{t,max}_{\xi_q}(s) = max_a \left\{ v^{t,max}_{\xi_q}(s, a) \right\}$$

An action $a'$ is pruned if there exists the same action $a$ over all samples $\xi_q$, such that

$$v^{t,min}_{\xi_q}(s, a) \geq v^{t,max}_{\xi_q}(s, a') \; \exists a, \; \forall \xi_q$$

The above pruning step follows from the observation that an action whose best case payoff is less than the worst case payoff of another action $a$ cannot be part of the regret optimal strategy, since we could switch from $a'$ to $a$ without increasing the regret value. It should be noted that an action that is not optimal for any of the samples cannot be pruned.

**Greedy sampling**

The scalability of the MILP formulation above is constrained by the number of samples $Q$. So, instead of generating only the fixed set of $Q$ samples from the uncertainty distribution over models, we generate more than $Q$ samples and then pick a set of size $Q$ so that samples are "as far apart" as possible. The key intuition in selecting the samples is to consider distance among samples as being equivalent to entropy in the optimal policies for the MDPs in the samples. For each decision epoch, $t$, each state $s$ and action $a$, we define $Pr^{s,a,t}_{\xi}(\pi^{*t}_{\xi}(s, a) = 1)$ to be the probability that $a$ is the optimal action in state $s$ at time $t$. Similarly, we define $Pr^{s,a,t}_{\xi}(\pi^{*t}_{\xi}(s, a) = 0)$:

$$Pr^{s,a,t}_{\xi}(\pi^{*t}_{\xi}(s, a) = 1) = \frac{\sum_{\xi_q} \pi^{*t}_{\xi_q}(s, a)}{Q}; Pr^{s,a,t}_{\xi}(\pi^{*t}_{\xi}(s, a) = 0) = \frac{\sum_{\xi_q} \left( 1 - \pi^{*t}_{\xi_q}(s, a) \right)}{Q}$$

Let the total entropy of sample set, $\xi$ ($|\xi| = Q$) be represented as $\Delta S(\xi)$, then

$$\Delta S(\xi) = - \sum_{t,s,a} \sum_{z \in \{0,1\}} Pr^{s,a,t}_{\xi}(\pi^{*t}_{\xi}(s, a) = z) \cdot ln\left( Pr^{s,a,t}_{\xi}(\pi^{*t}_{\xi}(s, a) = z) \right)$$

We use a greedy strategy to select the $Q$ samples, i.e. we iteratively add samples that maximize entropy of the sample set in that iteration.

It is possible to provide bounds on the number of samples required for a given error using the methods suggested by Shapiro *et al.* [14]. However these bounds are conservative and as we show in the experimental results section, typically, we only require a small number of samples.

## CER Minimizing Solution

The MILP based approach mentioned in the previous section can easily be adapted to minimize the maximum cumulative regret over all samples when uncertainties across states are dependent:

$$\min_{\vec{\pi}^0} \; creg(\vec{\pi}^0)$$

$$\textbf{s.t.} \;\; creg(\vec{\pi}^0) \geq \sum_s \alpha(s) \cdot creg^0_{\xi_q}(s, \vec{\pi}^t), \qquad\qquad\qquad\qquad \forall \xi_q$$

$$creg^t_{\xi_q}(s, \vec{\pi}^t) = \sum_a \pi^t(s,a) \cdot creg^t_{\xi_q}(s, a, \vec{\pi}^t), \qquad\qquad\qquad \forall s, t, \xi_q \quad (10)$$

$$creg^t_{\xi_q}(s, a, \vec{\pi}^t) = \mathcal{R}^{*,t}_q(s) - \mathcal{R}^t_q(s,a) + \gamma \sum_{s'} \mathcal{T}^t_q(s,a,s') \cdot creg^{t+1}_{\xi_q}(s', \vec{\pi}^{t+1}), \quad \forall s, a, t, \xi_q \quad (11)$$

where the product term $\pi^t(s,a) \cdot creg^t_{\xi_q}(s, a, \vec{\pi}^t)$ is approximated as described earlier.

While we were unable to exploit the independence of uncertainty distributions across states with *minimax* regret, we are able to exploit the independence with *minimax* CER. In fact, a key advantage of the CER robustness concept in the context of independent uncertainties is that it has the *optimal substructure* over time steps and hence a Dynamic Programming(DP) algorithm can be used to solve it.

In the case of independent uncertainties, samples at each time step can be drawn independently and we now introduce a formal notation to account for samples drawn at each time step. Let $\xi^t$ denote the set of samples at time step $t$, then $\xi = \times_{t \leq H-1} \xi^t$. Further, we use $\vec{\xi}^t$ to indicate cross product of samples from $t$ to $H-1$, i.e. $\vec{\xi}^t = \times_{t \leq e \leq H-1} \xi^e$. Thus, $\vec{\xi}^0 = \xi$. To indicate the entire horizon samples corresponding to a sample $p$ from time step $t$, we have $\vec{\xi}^t_p = \xi^t_p \times \vec{\xi}^{t+1}$.

For notational compactness, we use $\Delta \mathcal{R}^{t-1}_p(s,a) = \mathcal{R}^{*,t-1}_p(s) - \mathcal{R}^{t-1}_p(s,a)$. Because of independence in uncertainties across time steps, for a sample set $\vec{\xi}^{t-1}_p = \xi^{t-1}_p \times \vec{\xi}^t$, we have the following:

$$\max_{\vec{\xi}^{t-1}_p} creg^{t-1}_{\vec{\xi}^{t-1}_p}(s, \vec{\pi}^{t-1}) = \max_{\xi^{t-1}_p \times \xi^t_p} \sum_a \pi^{t-1}(s,a) \Big[ \Delta \mathcal{R}^{t-1}_p(s,a) + \gamma \sum_{s'} \mathcal{T}^t_p(s,a,s') \cdot creg^t_{\vec{\xi}^t}(s', \vec{\pi}^t) \Big]$$

$$= \max_{\xi^{t-1}_p} \sum_a \pi^{t-1}(s,a) \Big[ \Delta \mathcal{R}^{t-1}_p(s,a) + \gamma \sum_{s'} \mathcal{T}^t_p(s,a,s') \cdot \max_{\vec{\xi}^t_q \in \vec{\xi}^t} creg^t_{\vec{\xi}^t_q}(s', \vec{\pi}^t) \Big] \tag{12}$$

**Proposition 4** *At time step $t-1$, the CER corresponding to any policy $\pi^{t-1}$ will have least regret if it includes the CER minimizing policy from $t$. Formally, if $\vec{\pi}^{*,t}$ represents the CER minimizing policy from $t$ and $\vec{\pi}^t$ represents any arbitrary policy, then:*

$$\forall s : \max_{\vec{\xi}^{t-1}_p \in \vec{\xi}^{t-1}} creg^{t-1}_{\vec{\xi}^{t-1}_p}\Big(s, \langle \pi^{t-1}, \vec{\pi}^{*,t} \rangle\Big) \leq \max_{\vec{\xi}^{t-1}_p \in \vec{\xi}^{t-1}} creg^{t-1}_{\vec{\xi}^{t-1}_p}\Big(s, \langle \pi^{t-1}, \vec{\pi}^t \rangle\Big) \tag{13}$$

$$\textbf{\textit{if}}, \;\; \forall s : \max_{\vec{\xi}^t_q \in \vec{\xi}^t} creg^t_{\vec{\xi}^t_q}(s, \vec{\pi}^{*,t}) \leq \max_{\vec{\xi}^t_q \in \vec{\xi}^t} creg^t_{\vec{\xi}^t_q}(s, \vec{\pi}^t) \tag{14}$$

**Proof Sketch**[6] We prove this by using Equations (14) and (12) in LHS of Equation (13). ∎

It is easy to show that minimizing CER also has an optimal substructure:

$$\min_{\vec{\pi}^0} \max_{\vec{\xi}^0_p} \sum_s \alpha(s) \cdot creg^0_{\vec{\xi}^0_p}(s, \vec{\pi}^0) \implies \min_{\vec{\pi}^0} \sum_s \alpha(s) \cdot \Big[ \max_{\vec{\xi}^0_p} creg^0_{\vec{\xi}^0_p}(s, \vec{\pi}^0) \Big] \tag{15}$$

In Proposition 4 (extending the reasoning to $t = 1$), we have already shown that $\max_{\vec{\xi}^0_p} creg^0_{\vec{\xi}^0_p}(s, \vec{\pi}^0)$ has an optimal substructure. Thus, Equation (15) can also exploit the optimal substructure.

MINIMIZECER function below provides the pseudo code for a DP algorithm that exploits this structure. At each stage, $t$ we calculate the $creg$ for each state-action pair corresponding to each of the

samples at that stage, i.e. $\xi^t$ (lines 6-9). Once these are computed, we obtain the maximum $creg$ and the policy corresponding to it (line 10) using the GETCER() function. In the next iteration, $creg$ computed at $t$ is then used in the computation of $creg$ at $t-1$ using the same update step (lines 6-9).

---

MINIMIZECER()

1: **for all** $t \leq H - 1$ **do**
2: $\quad \xi^t \leftarrow$ GENSAMPLES($\mathbf{T}, \mathbf{R}$)
3: **for all** $s \in \mathcal{S}$ **do**
4: $\quad creg^H(s) \leftarrow 0$
5: **while** $t >= 0$ **do**
6: $\quad$ **for all** $s \in \mathcal{S}$ **do**
7: $\quad\quad$ **for all** $\xi_q^t \in \xi^t, a \in \mathcal{A}$ **do**
8: $\quad\quad\quad creg_{\xi_q^t}^t(s,a) \leftarrow \Delta R_q^t(s,a) +$
9: $\quad\quad\quad\quad \gamma \sum_{s'} \mathcal{T}_q^t(s,a,s') \cdot creg^{t+1}(s')$
10: $\quad\quad \langle \pi^t, creg^t(s) \rangle \leftarrow$ GETCER (s, $\{creg_{\xi_q^t}^t(s,a)\}$)
11: $\quad t \leftarrow t - 1$
$\quad$ **return** $(\vec{creg}^0, \vec{\pi}^0)$

GETCER $(s, \{creg_{\xi_q^t}^t(s,a)\})$

$$\min_{\boldsymbol{\pi}} creg^t(s)$$

$$creg^t(s) \geq \sum_a \pi^t(s,a) \cdot creg_{\xi_q^t}^t(s,a), \; \forall \xi_q^t$$

$$\sum_a \pi^t(s,a) = 1$$

$$0 \leq \pi^t(s,a) \leq 1, \forall a$$

---

It can be noted that MINIMIZECER() makes only $H \cdot |\mathcal{S}|$ calls to the LP in GETCER() function, each of which has only $|\mathcal{A}|$ continuous variables and at most $[1 + \max_t |\xi^t|]$ number of constraints. Thus, the overall complexity of MinimizeCER() is polynomial in the number of samples given fixed values of other attributes.

Let $creg^{*,H-1}(s,a)$ denote the optimal cumulative regret at time step $H-1$ for taking action $a$ in state $s$ and $creg_\xi^{*,H-1}(s,a)$ denote the optimal cumulative regret over the sample set $\xi$. Let indicator random variable, $X$ be defined as follows: $X = \begin{cases} 1 & \text{if } creg^{*,H-1}(s,a) - creg_\xi^{*,H-1}(s,a) \leq \lambda \\ 0 & otherwise \end{cases}$

By using Chernoff and Hoeffding bounds on $X$, it is possible to provide bounds on deviation from mean and on the number of samples at $H-1$. This can then be propagated to $H-2$ and so on. However, these bounds can be very loose and they do not exploit the properties of $creg$ functions. Bounds developed on spacings of order statistics can help exploit the properties of $creg$ functions. We will leave this for future work.

## Experimental Results

In this section, we provide performance comparison of various algorithms introduced in previous sections over two domains. MILP-Regret refers to the randomized policy variant of the MILP approximation algorithm for solving uncertain MDPs with dependent uncertainties. Similar one for minimizing CER is referred to as MILP-CER. We refer to the dynamic programming algorithm for minimizing CER in the independent uncertainty case as DP-CER and finally, we refer to the *maximin* value algorithm as "Maximin". All the algorithms finished within 15 minutes on all the problems. DP-CER was much faster than other algorithms and finished within a minute on the largest problems.

We provide the following results in this section:

(1) Performance comparison of Greedy sampling and Random sampling strategies in the context of MILP-Regret as we increase the number of samples.
(2) SAA analysis of the results obtained using MILP-Regret.
(3) Comparison of MILP-Regret and MILP-CER policies with respect to simulated regret.
(4) Comparison of DP-CER and Maximin.

The first three comparisons correspond to the dependent uncertainties case and the results are based on a path planning problem that is motivated by disaster rescue and is a modification of the one employed in Bagnell *et al.* [1]. On top of normal transitional uncertainty, we have uncertainty over transition and reward models due to random obstacles and random reward cells. Furthermore, these uncertainties are dependent on each other due to patterns in terrains. Each sample of the various uncertainties represents an individual map and can be modelled as an MDP. We experimented with

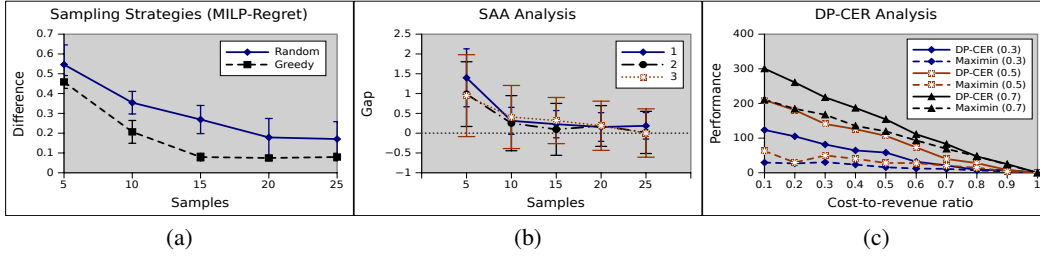

Figure 1: In (a),(b) we have $4 \times 4$ grid, $H = 5$. In (c), the maximum inventory size $(X) = 50$, $H = 20$, $|\xi^t| = 50$. The normal distribution mean $\mu = \{0.3, 0.4, 0.5\} \cdot X$ and $\sigma \leq \frac{min\{\mu, X-\mu\}}{3}$

a grid world of size 4x4 while varying numbers of obstacles, reward cells, horizon and the number of break points employed (3-6).

In Figure 1a, we show the effect of using greedy sampling strategy on the MILP-Regret policy. On X-axis, we represent the number of samples used for computation of policy (learning set). The test set from which the samples were selected consisted of 250 samples. We then obtained the policies using MILP-Regret corresponding to the sample sets (referred to as learning set) generated by using the two sampling strategies. On Y-axis, we show the percentage difference between simulated regret values on test and learning sample sets. We observe that for a fixed difference, the number of samples required by greedy is significantly lower in comparison to random. Furthermore, the variance in difference is also much lower for greedy. A key result from this graph is that even with just 15 samples, the difference with actual regret is less than 10%.

Figure 1b shows that even the gap obtained using SAA analysis[7] is near zero ($< 0.1$) with 15 samples. We have shown the gap and variance on the gap over three different settings of uncertainty labeled 1,2 and 3. Setting 3 has the highest uncertainty over the models and Setting 1 has the least uncertainty. The variance over the gap was higher for higher uncertainty settings.

While MILP-CER obtained a simulated regret value (over 250 samples) within the bound provided in Proposition 1, we were unable to find any correlation in the simulated regret values of MILP-Regret and MILP-CER policies as the samples were increased. We have not yet ascertained a reason for there being no correlation in performance.

In the last result shown in Figure 1c, we employ the well known single product finite horizon stochastic inventory control problem [10]. We compare DP-CER against the widely used benchmark algorithm on this domain, Maximin. The demand values at each decision epoch were taken from a normal distribution. We considered three different settings of mean and variance of the demand. As expected, the DP-CER approach provides much higher values than maximin and the difference between the two reduced as the cost to revenue ratio increased. We obtained similar results when the demands were taken from other distributions (uniform and bi-modal).

## Conclusions

We have introduced scalable sampling-based mechanisms for optimizing regret and a new variant of regret called CER in uncertain MDPs with dependent and independent uncertainties across states. We have provided a variety of theoretical results that indicate the connection between regret and CER, quality bounds on regret in case of MILP-Regret, optimal substructure in optimizing CER for independent uncertainty case and run-time performance for MinimizeCER. In the future, we hope to better understand the correlation between regret and CER, while also understanding the properties of CER policies.

**Acknowledgement** This research was supported in part by the National Research Foundation Singapore through the Singapore MIT Alliance for Research and Technologys Future Urban Mobility research programme. The last author was also supported by ONR grant N00014-12-1-0999.

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

## Footnotes

[1]Detailed proof provided in supplement under Proposition 1.

[2]Using CPLEX SOS-2 considerably improves runtime compared to a binary variables formulation.

[3]Proposition and proof provided in supplement as footnote 3

[4]Detailed proof in supplement under Proposition 3

[5]Pseudo code provided in the supplement under "Pruning dominated actions" section.

[6]Detailed proof in supplement under Proposition 4.

[7]We have provided the method for performing SAA analysis in the supplement.
