[Supplementary Material]

# Regret based Solutions for Uncertain MDPs

**Proposition 1.** *For a policy* $\vec{\pi}^0 : 0 \le reg(\vec{\pi}^0) - creg(\vec{\pi}^0) \le \left[ \max_s R^*(s) - \min_s R^*(s) \right] \cdot \frac{(1-\gamma^H)}{1-\gamma}$

**Proof.** We can rewrite Equation (1) as follows:

$$creg(\vec{\pi}^0) = v^{0,\#}(\vec{\pi}^0) - v^0(\vec{\pi}^0), \textbf{ where } v^{0,\#}(\vec{\pi}^0) = \sum_s \alpha(s)v^{0,\#}(s,\vec{\pi}^0) \textbf{ and }$$

$$v^{t,\#}(s,\vec{\pi}^t) = \sum_a \pi^t(s,a) \cdot \left[ R^*(s) + \gamma \sum_{s'} T(s,a,s') \cdot v^{t+1,\#}(s',\vec{\pi}^{t+1}) \right]$$

Since the value for any policy cannot exceed the value of optimal policy, we have:
$reg(\vec{\pi}^0) - creg(\vec{\pi}^0) = v^0(\vec{\pi}^*) - v^{0,\#}(\vec{\pi}^0) \ge 0$. The difference in value of optimal policy (a deterministic one) and any other policy is because of the states visited by using the policy. In the worst case for $creg$, the optimal policy visits the state with highest $R^*$ and $\vec{\pi}^0$ visits the states with the lowest $R^*$ at every time step. Sum of a geometric progression over the time steps yields

$$reg(\vec{\pi}^0) - creg(\vec{\pi}^0) \le \left[ \max_s R^*(s) - \min_s R^*(s) \right] \cdot \frac{(1-\gamma^H)}{1-\gamma}$$

(a)

Figure 1: Error

The proposition below provides the proof for footnote 2.

**Footnote 3.** *In the approximation of $x^2$ function using piecewise linear components, $\lambda(w)$, the maximum approximation in any interval $[br_{w-1}, br_w]$ occurs at the mid-point.*

**Proof.** Without loss of generality, let us consider any point $y$ in the interval $[br_{w-1}, br_w]$. From Equation 7, we have

$$y = \lambda_{w-1}br_{w-1} + \lambda_w br_w$$

Since, we have the sum constraint in Equation 9, the above equation can be modified as:

$$y = (1 - \lambda_w)br_{w-1} + \lambda_w br_w$$

$$\implies \lambda_w = \frac{y - br_{w-1}}{br_w - br_{w-1}}$$

The error is given by the difference between LHS and RHS in Equation 8:

$$\delta = y^2 - \left[\lambda_{w-1}(br_{w-1})^2 + \lambda_w(br_w)^2\right]$$

Substituting value of $\lambda_w$:

$$\delta = y^2 - \left[(br_w)^2 \cdot \frac{y - br_{w-1}}{br_w - br_{w-1}} - (br_{w-1})^2 \cdot \frac{y - br_w}{br_w - br_{w-1}}\right]$$

When $\delta$ is maximum, we have $\frac{d\delta}{dy} = 0$. Therefore:

$$2y - \frac{((br_w)^2 - (br_{w-1})^2)}{(br_w - br_{w-1})} = 0$$

$$y = \frac{br_w + br_{w-1}}{2}$$

Hence proved. ∎

**Proposition 3.** *Let $\hat{v}^t_{\xi_q}(s, \vec{\pi}^t)$ denote the approximation of $v^t_{\xi_q}(s, \vec{\pi}^t)$. Then*

$$v^t_{\xi_q}(s, \vec{\pi}^t) - \frac{|\mathcal{A}| \cdot \epsilon \cdot (1 - \gamma^{H-1})}{4 \cdot (1 - \gamma)} \leq \hat{v}^t_{\xi_q}(s, \vec{\pi}^t) \leq v^t_{\xi_q}(s, \vec{\pi}^t) + \frac{|\mathcal{A}| \cdot \epsilon \cdot (1 - \gamma^{H-1})}{4 \cdot (1 - \gamma)}$$

**Proof:** At time step $t + 1$, the approximation error in $v^{t+1}_{\xi_q}(s, \vec{\pi}^{t+1})$ is given by $|\mathcal{A}| \cdot \delta$, ($|\mathcal{A}|$ as maximum number of actions across all states, all time steps). The maximum approximation error at time step $t$ in $v^t_{\xi_q}(s, a, \vec{\pi}^t)$ is $\gamma \cdot |\mathcal{A}| \cdot \delta$ (due to error in value function at time step $t + 1$). We can combine equation 3 and 4 as:

$$v^t_{\xi_q}(s, \vec{\pi}^t) = \sum_a \pi^t(s, a) \cdot \left[v^t_{\xi_q}(s, a, \vec{\pi}^t) \pm \gamma \cdot |\mathcal{A}| \cdot \delta\right]$$

$$= \sum_a \pi^t(s, a) \cdot v^t_{\xi_q}(s, a, \vec{\pi}^t) \pm \gamma \cdot |\mathcal{A}| \cdot \delta$$

Now at time step $t$ the error will be $|\mathcal{A}| \cdot \delta$ plus future error from time step $t + 1$ given by $\gamma \cdot |\mathcal{A}| \cdot \delta$. Extending to $t = 0$ we will have sum of two geometric progressions, i.e.

$$\pm \left[|\mathcal{A}| \cdot \delta + \gamma \cdot |\mathcal{A}| \cdot \delta + \gamma^2 \cdot |\mathcal{A}| \cdot \delta ... \right]$$

Substituting $\delta = \frac{\epsilon}{4}$, we will have a positive and negative error of $\frac{|\mathcal{A}| \cdot \epsilon \cdot (1 - \gamma^{H-1})}{4 \cdot (1 - \gamma)}$. ∎

**Proposition 4.** *At time step $t - 1$, the CER corresponding to any policy, $\pi^{t-1}$ will have least regret if it includes CER minimizing policy from $t$. Formally, if $\vec{\pi}^{*,t}$ represents the CER minimizing policy from $t$ and $\vec{\pi}^t$ represents any arbitrary policy, then:*

$$\forall s: \max_{\vec{\xi}^{t-1}_p \in \vec{\xi}^{t-1}} creg^{t-1}_{\vec{\xi}^{t-1}_p}\left(s, \langle \pi^{t-1}, \vec{\pi}^{*,t} \rangle\right) \leq \max_{\vec{\xi}^{t-1}_p \in \vec{\xi}^{t-1}} creg^{t-1}_{\vec{\xi}^{t-1}_p}\left(s, \langle \pi^{t-1}, \vec{\pi}^t \rangle\right)$$

$$\mathbf{\textit{if}}, \ \forall s: \max_{\vec{\xi}^t_q \in \vec{\xi}^t} creg^t_{\vec{\xi}^t_q}(s, \vec{\pi}^{*,t}) \leq \max_{\vec{\xi}^t_q \in \vec{\xi}^t} creg^t_{\vec{\xi}^t_q}(s, \vec{\pi}^t)$$

**Proof.** From Equation 12, we have:

$$creg^{t-1}_{\vec{\xi}^{t-1}_p}\left(s, \langle \pi^{t-1}, \vec{\pi}^{*,t} \rangle\right) = \sum_{a \in \mathcal{A}} \pi^{t-1}(s, a)\left[\Delta \mathcal{R}^{t-1}_p(s, a) + \gamma \sum_{s'} \mathcal{T}^{t-1}_p(s, a, s') \cdot \max_{\vec{\xi}^t_q \in \vec{\xi}^t} creg^t_{\vec{\xi}^t_q}(s', \vec{\pi}^{*,t})\right]$$

From Equation 14, we have:

$$creg^{t-1}_{\vec{\xi}^{t-1}_p}\left(s, \langle \pi^{t-1}, \vec{\pi}^{*,t} \rangle\right) \leq \sum_{a \in \mathcal{A}} \pi^{t-1}(s, a)\left[\Delta \mathcal{R}^{t-1}_p(s, a) + \gamma \sum_{s'} \mathcal{T}^{t-1}_p(s, a, s') \cdot \max_{\vec{\xi}^t_q \in \vec{\xi}^t} creg^t_{\vec{\xi}^t_q}(s', \vec{\pi}^t)\right]$$

$$\leq creg^{t-1}_{\vec{\xi}^{t-1}_p}\left(s, \langle \pi^{t-1}, \vec{\pi}^t \rangle\right)$$

$$\mathbf{\textit{Thus}}, \ \max_{\xi_q \in \xi} creg^{t-1}_{\xi_q}\left(s, \langle \pi^{t-1}, \vec{\pi}^{*,t} \rangle\right) \leq \max_{\xi_q \in \xi} creg^{t-1}_{\xi_q}\left(s, \langle \pi^{t-1}, \vec{\pi}^t \rangle\right). \ \blacksquare$$

## Pruning dominated actions

Algorithm 1 provides the pseudo-code for pruning step discussed earlier. At each time step, for each state we maintain an upper and lower bound for the value function. Apart from pruning, this gives us tight bounds on value function that decrease the number of break points required for linearization.

---

**Algorithm 1:** PRUNEDOMINATEDACTIONS()

---

$t \leftarrow H - 1$
**for all** $\xi_q \in \xi, s \in \mathcal{S}$ **do**
  $v_{\xi_q}^{H,min}(s) \leftarrow 0$
  $v_{\xi_q}^{H,max}(s) \leftarrow 0$
**while** $t >= 0$ **do**
  **for all** $s \in \mathcal{S}$ **do**
    **for all** $\xi_q \in \xi, a \in \mathcal{A}$ **do**
      $v_{\xi_q}^{t,min}(s,a) \leftarrow R_q^t(s,a) + \gamma \sum_{s'} \mathcal{T}_q^t(s,a,s') \cdot v_{\xi_q}^{t+1,min}(s')$
      $v_{\xi_q}^{t,max}(s,a) \leftarrow R_q^t(s,a) + \gamma \sum_{s'} \mathcal{T}_q^t(s,a,s') \cdot v_{\xi_q}^{t+1,max}(s')$
    **if** $\exists a' \ s.t. \ v_{\xi_q}^{t,min}(s,a') \geq v_{\xi_q}^{t,max}(s,a) \ \forall \xi_q$ **then**
      PRUNE $a$
    $v_{\xi_q}^{t+1,min}(s) = min_a v_{\xi_q}^{t,min}(s,a)$
    $v_{\xi_q}^{t+1,max}(s) = max_a v_{\xi_q}^{t,max}(s,a)$
  $t \leftarrow t - 1$

---

## SAA Analysis

Each sample (scenario) is described by $i = \{i_1, i_2, i_3, ..., i_{|T|}\}$ and belong to the set $I$ (in the case where we consider independent transition probabilities/rewards in each stage, $I$ is the set of samples which are cross products of independent samples in each stage). Followed from the sample average approximation (SAA) method described by [2], the steps to calculate the approximate optimality gap are as follows:

1. Generate the set of sample sets, $M = \{I_1, I_2, ..., I_{|M|}\}$, where each sample set is of size $|I|$. Also generate a larger sample set of size $|I'| \gg |I|$.

   - For $m = 1, ..., |M|$, solve the problem with sample set $I_m$ to obtain the solution value $r\bar{e}g_m^*$ and policy $\bar{\pi}_m$

2. Compute the average of the objective values obtained which is a statistical lower bound of the problem and their corresponding variance as follows:

$$r\hat{e}g^* = \frac{1}{|M|} \sum_{m \in M} r\bar{e}g_m^* \text{ and } \sigma_{r\hat{e}g^*}^2 = \frac{1}{|M|(|M|-1)} \sum_{m \in M} (r\bar{e}g_m^* - r\hat{e}g^*)^2 .$$

3. Let $\bar{\pi}$ be the selected solution from the set of solutions obtained in Step 1. Denote by $reg_{I'}^*(\bar{\pi})$ the regret value of the policy $\bar{\pi}$ on the large sample set $I'$. This value is the sample average estimate of the true objective function of the policy $\bar{\pi}$. Also, its variance can be computed as follows:

$$\sigma_{I'}^2(\bar{\pi}) = \frac{1}{|I'|(|I'|-1)} \sum_{i \in I'} (r\bar{e}g_i^*(\bar{\pi}) - reg_{I'}^*(\bar{\pi}))^2$$

   where $r\bar{e}g_i^*(\bar{\pi})$ is the regret of the policy $\bar{\pi}$ corresponding to each sample $i \in I'$.

4. The absolute optimality gap of the solution $\bar{\pi}$ and its variance can be estimated as follows:

$$gap(\bar{\pi}) = |reg_{I'}^*(\bar{\pi}) - r\hat{e}g^*| \text{ and } \sigma_{gap}^2(\bar{\pi}) = \sigma_{I'}^2(\bar{\pi}) + \sigma_{r\hat{e}g^*}^2 .$$

We can similarly perform SAA analysis for MILP-CER.

## Single Product Stochastic Inventory Control Problem

In the single product finite horizon stochastic inventory control problem [1], at the beginning of each time period and before observing the demand, the manager determines the current inventory size $x^t$ and decides whether or not to order additional stock $y^t$ from a supplier. We assume the cost of ordering $u$ units is given by $k_1 \cdot u$, the cost of maintaining an inventory of $u$ units is given by $k_2 \cdot u$ and the revenue obtained when the demand is $j$ units is given by $k_3 \cdot j$.

Denote $D^t = \left\{ d_0^t, d_1^t, ..., d_q^t \right\}$ as the set of demand values at time step $t$ (independent of demand in other time steps). The inventory at time step $t+1$ for demand $d_q^t$ is given by $x^{t+1}(d_q^t) = max \left\{ x^t + y^t - d_q^t, 0 \right\} \equiv [x^t + y^t - d_q^t]^+$. Note the that reward at time step $t$ depends on the current and subsequent inventory size and is given by $r^t(x^t, y^t, x^{t+1}(d_q^t)) = -k_1 \cdot y^t - k_2 \cdot (x^t + y^t) + k_3 \cdot ([x^t + y^t - x^{t+1}(d_q^t)]^+)$.

The discrete demand uncertainty values translate to uncertainty over reward and transition functions, which require robust solution concepts. A standard approach is to maximize the minimum expected values or *maximin solution*. In this paper, we compare DP-CER against maximin across different cost-to-revenue ratio defined as $\frac{k_1+k_2}{k_3}$.