[Reviews · NeurIPS 2013]

Submitted by Assigned_Reviewer_4

The paper presents a number of contributions towards the solution of uncertain MDPs with minimax regret. The paper introduces methods to handle uncertainty over transition dynamics, something that has not been presented before. The paper also discusses handling of dependencies of the model distributions across states andactions and decision epochs. The paper is very well written and well motivated. The results, both theoretical and empirical, look good. I have not looked at the supplementary material, but find the paper does a good job of relegating the right level of detail therein.

Only two suggestions for improvement, mainly to improve the readabiliy for researchers outside of the direct area of this work
- First, it would be good to have some explanation for why the minimax regret outperforms the maximin solutions. Just prior to concluding, the paper states that "As expected, the DP-CER approach [outperforms] maximin...", but I could not find where this "expectation" is expressed in the paper. Why does one expect this?
- Second, can you provide some intuition about the invertory control problem's model distributions - are they independent according to definition 3, or just treated that way by DP-CER. Does the maximin algorithm also treat them this way, or does it model the dependencies?


A few minor comments
- there is an extra "dot" before \alpha in the first equation in Definition 2
- last page of text - referring to fig. 1(b) the text says that just 25 samples are needed for 10% regret difference, but it seems that for the greedy approach this is only 15 samples?
- the third to last paragraph before the conclusion begins with "Our result (2) shows that ..." - it is unclear what (2) refers to here
Summary: A good paper presenting new results for minimax regret-based solutions to uncertain MDPs.

Submitted by Assigned_Reviewer_5

Title: Regret based Robust Solutions for Uncertain Markov Decision Processes

Summary: This paper proposes algorithms for computing robust solutions for acting in unknown MDPs in an off-line mode, where a given uncertain MDP is provided. Within this setting, the main contribution of the paper is to propose algorithms that can handle uncertainty on the transition dynamics. The algorithms are based on the formalization of the regret minimization problem as Mixed Integer Linear Program (MILP). Both empirical and experimental results are provided.

My first understanding of the paper was wrong, and I really apologize for it. The authors's feedback helped me better understanding the contribution of the paper. In particular, I did not realize that the uncertain MDP was fully known, and that the uncertainty was not vanishing over time.


Minor comments:

- line 048: "Chernoff and Hoeffiding" bounds -> Chernoff-Hoeffding bounds;
- line 065: "$\alpha(s)$ to denote the starting state distribution in state $s$ -> what does "in state $s$" means? I guess that, in the context of a finite state space, $\alpha(s)$ is the probability of starting from $s$;
- "markov" should take a capital letter "Markov" in the references;
- dots for signifying products would be erased;
- the arrow on top of policies does not seems useful;
- line 087: the $\#$ is undefined;
- line 089: Corollary 1 studies a specific case for which optimal rewards are the same in every state. This corresponds to a case where the greedy policy with respect to the reward function is optimal. This looks strange;
- line 099: "according to the $k$th element in $\mathcal T$". Does it mean that $\mathcal T$ is a countable set? This looks strange. I have the same remark for $\mathcal R$.
Summary: This paper proposes new robust-type algorithms for solving uncertain MDPs. In particular, such algorithms can handle uncertainty on the transitions dynamics.

Submitted by Assigned_Reviewer_6

This paper makes several good contributions to the literature on robust MDPs. In contrast to previous work, this paper handles both reward and transition uncertainty and also breaks away from making strong assumptions on the underlying uncertainty distributions.

One innovation is to sample MDPs from the underlying uncertainty distribution to avoid having to represent the distributions explicitly (as say, a Gaussian reward distribution). They employ MILP to exactly solve for a single deterministic (history-dependent?) policy that minimizes the max regret attainable in a sample of |\xi| MDPs. Or they can approximate a randomized policy for the same problem and the approximation is a bounded distance away from the actual solution [Propositions 2 and 3].

Another innovation is to adapt Cumulative Expected Regret (CER) from the multi-armed bandits literature and apply it specifically for MDPs, which if they also assume independence of uncertainty distributions across states / actions / epoch, then they can scale up to solve uncertain MDP problems with a longish horizon (H = 20) [Proposition 4]. But if one can not assume the independence of uncertainty distributions, then the scale up isn't possible?

These contributions are interesting and worthy of publication.

I'm slightly unclear on a few things to do with the MDP problem formulation. On line 063, the way that policy is written looks like we're potentially exploring the space of history-dependent policies. But possibly it could mean that it's just dependent on the time horizon we're in. If it's the former, then is 'state' not simply the current state, but rather entire rolling histories of state (e.g. state at time 2 = {low in time 0, high in time 1, low in time 2} and not state in time 2= {low in time 2})? And if it is rolling histories of state, what implications does that have for proposition and corollary 1?

There's another paper that also takes a sampling-based approach to handling uncertain MDPs from NIPS 2012 called "Tractable Objectives for Robust Policy Optimization". It may be worthwhile to think about how the two papers are related even if that paper isn't looking at a minimax regret objetive.

Minor Comments:

- Hyphenate 'sampling-based' (e.g. line 050), 'regret-based' (e.g. line 051), state-action' (e.g. line 109).

- In a few places, there are parens where it might be cleaner to omit. For example:

- (line 064) R*(s) (=\max_a R(s,a)) could just be R*(s)=\max_a R(s,a)

- (line 098) s (\in S) could just be s \in S

- (line 099 s' (\in S) and a (\in A) can be s' \in S and a \in A.

- (line 206) \epsilon (= \frac{d^2 - c^2}{r}) can be \epsilon = \frac{d^2 - c^2}{r}

- There are some places where I'm not sure the italicizing maximin, reg, creg, SOS2 and br is necessary. Italicize when introducing a term for the first time but not anywhere else (e.g. for maximin). And in an equation, an italicized 'reg' (or 'SOS' or 'br') looks like r*e*g instead of like \text{reg}.

- The recursive equations need base-cases:

- (p. 2 Definition 1 and 2)

- (Supplemental proof of proposition 1)

- (p. 2 Preliminaries) you define t and H down on line 099 and 102 but you refer to them already in line 061.

- (p. 2 Preliminaries) Both the vanilla MDP and uncertain MDP are finite state, finite action, finite horizon problems. But only finite horizon is mentioned on line 096. Point this out by the vanilla MDP problem statement.

- (line 087) The '#' comes a bit out of nowhere as it is defined in the supplemental section and even there I'm not sure what it denotes.

- (p. 4 Equation 5) needs a qualifier that this for all s,a, t, \xi_q?

- (p 4, footnote 3) Footnote -> footnote

- (line 305) policy will have least regret if it includes *the* CER minimizing policy from t

- (line 362) and finally, we refer to the maximin value algorithm as 'Maximin' : if you're looking for stuff to cut to make room, I'd cut this.

- (Supplemental 025) *Summing* a geometric progression over the time steps yields

- (Supplemental line 134) belong-> belongs

- (Supplemental line 134) i = {i_1 … i_H} instead of i = {i_1 … i_{|T|}}?
Summary: The paper makes a significant contribution to planning in robust MDPs, by moving toward handling arbitrary reward and transition uncertainty.

Submitted by Assigned_Reviewer_7

This paper tackles the problem of solving MDPs with uncertainty in the parameters. The main contribution is to propose an optimization criteria based on minimizing the cumulative expected regret. The paper provides approximate algorithms for computing the solution efficiently under different assumptions, and a theoretical characterization of the cumulative expected regret under those assumptions. Empirical results are presented in 2 domains from the literature, showing better performance than the previously used Maximin criteria.

The paper contains a number of novel and interesting ideas in terms of solving uncertain MDPs. Relevant previous work is cited, and the paper clearly explains its new contributions. I think the definition of the CER criteria is worthwhile, as are the algorithmic approximations and theoretical analysis.

My main concern with the paper is that it is really not self-contained within the 8-pages. There are numerous references to the supplemental material, and as a reader, I found it really necessary to consult this material in a number of places (e.g. to understand the SAA analysis.) I'm not sure whether this is suitable for NIPS. On the one hand, I think the work presents interesting contributions. On the other hand, it really needs to be a longer paper to be solid. It seems a journal would be better suited for this work. I leave it to the chairs to weigh in on this one.

Detailed comments:
- l.62: Why do you consider a policy set, rather than a single policy?
- l.76: Can you give some intuition for creg()?
- l.88: What's the meaning of v^{0,#}?
- l.89: Corollary 1 seems trivial. When would you have a useful MDP with R*(s)=R*(s')? Is this result necessary? Or am I misunderstanding something?
- l.111: What is the meaning of "X" at the beginning of the RHS?
- l.253: Can you motivate the choice of entropy to measure distance among samples?
- l.405: "three different settings of uncertainty" -> Give enough detail that readers can reproduce the work.
- l.287: I did not understand the notion of substructure.
- l.418: What is the Y-axis for the Fig.1c?
Summary: A paper with interesting technical contributions, but not sufficiently self-contained.
Author Feedback

Author rebuttal: We thank all the reviewers for their valuable and detailed feedback. All the minor comments will be addressed directly in the paper.

Reviewer 4:

The maximin approach computes the best policy assuming the worst-case realization of the uncertainty and this is the main reason for maximin performing worse with respect to expected value in comparison with regret based approaches. Reviewer is right in pointing out the lack of substantiation in the sentence starting with “As expected”. We will include the intuition provided here and remove “As expected”.

For inventory control problems ([10]), the model distributions are independent according to definition [3]. The maximin approach is also able to exploit the independence in distributions.

Reviewer 5:

Unfortunately, the reviewer has misunderstood our contributions. While there are some minor similarities to the suggested papers, there are significant differences both in the problem model and in the objectives of solving the model.

Model: In our work, we do not assume or require a probability distribution over the unknown parameters as required in Bayesian RL. This is also one of the key differences with the Percentile Optimization paper by E. Delage and S. Mannor.

Objective in solving the model: Our work is to compute robust policies (that are based on regret). Standard Bayesian RL approaches compute expected reward maximizing policies. In Percentile Optimization work, the objective resembles a chance constraint. Such differing objectives imply different techniques.

UCRL2 provides regret bounds during learning and corresponds to online regret [Near-optimal regret bounds for RL, Auer et al, NIPS 2009. Logarithmic online regret bounds for RL, Peter Auer and Ronald Ortner, NIPS 2007]. Regret in our context is offline and for the computed policies over potential realizations of uncertainty.

Dependencies in our context correspond to dependence in uncertainty distributions in the given “model”. However, the references suggested by the reviewer correspond to using local accuracy for future exploration in the learning “algorithms”.

We will briefly summarize Bayesian RL in our related work.

The papers on Risk Aware Dynamic Programming consider traditional MDP models with risk objectives and hence are not directly relevant.

Reviewer 6:

Comment/question about the independence assumption in Proposition 4: If there is no independence, then scaling to big problems such as inventory control, while providing quality guarantees is indeed a challenge and is not addressed in our paper.

The policy definition in 063 corresponds to the latter interpretation, i.e. it is dependent on the time horizon we’re in. At each time step, we have a mixed strategy. Considering policies with rolling histories of states would be an interesting and significant challenge for future work.

The suggested paper on tractable objectives is relevant and we will be including it in our related work. It would indeed be an interesting exercise to compare our tractable objective CER with the tractable objectives provided by [Katherine and Bowling].


Reviewer 7:

While this is a subjective issue, we wished to indicate that SAA analysis is a very well known topic in Stochastic Optimization and hence we have included it in supplementary material. Leaving that aside, like many other previous NIPS papers (http://books.nips.cc/nips24.html), we have only included detailed proofs for a few of the propositions (for which proof sketches are provided in main paper) and pseudocode (intuition for algorithm provided in main paper) in supplementary material.

We will fix the minor comments regarding missing definitions of symbols and descriptions/intuitions in the paper.

1.62: Ours is a finite horizon problem and hence we provide a mapping from states to probability distribution over actions at each time step. We denote this using the set representation.

1.76: Intuitively, creg can be considered as accumulation of expected regrets at each time step. Informally, it would be a myopic variant of traditional regret.

1.88: v^{0,#} is an intermediate symbol, which represents upper bound on expected value obtained by transitioning according to actions of the given policy.

1.89: While trivial, the goal with Corollary 1 was to provide a condition where creg and reg would be equal.

1.111: It is “times” or cross product. In that equation, it represents the cross product of uncertainty distributions over all states, actions and decision epochs.

1.253: In the context of sampling, distance between two samples represents the difference in optimal policies of two samples. If two samples are close, the optimal policies are similar. We chose entropy for computing similarity in policies, because it is directly proportional to distance between samples. If two samples are close (distance wise), the optimal policies are similar and hence the expression in 1.261 for entropy returns a low value. On the other hand, if two samples are far, optimal policies are dissimilar and hence entropy is high.

1.287: Optimal substructure is a property where in optimal solutions can be constructed efficiently from optimal solutions to subproblems. In our case, the structure implies that optimal creg policy for a higher horizon problem can be computed from optimal policy for lower horizon problem.

1.418: Y-axis represents expected value.